# Cytokine Receptor-like Factor 1 (CRLF1) and Its Role in Osteochondral Repair

**DOI:** 10.3390/cells13090757

**Published:** 2024-04-28

**Authors:** Fenglin Zhang, Andrew J. Clair, John F. Dankert, You Jin Lee, Kirk A. Campbell, Thorsten Kirsch

**Affiliations:** 1Department of Urology, New York University Grossman School of Medicine, New York, NY 10010, USA; fenglin.zhang@nyulangone.org; 2Rothman Orthopedic Institute, Orlando, FL 32803, USA; andrew.clair@rothmanortho.com; 3Department of Orthopedic Surgery, New York University Grossman School of Medicine, New York, NY 10010, USA; john.dankert@nyulangone.org (J.F.D.); eugene.lee@nyulangone.org (Y.J.L.); kirk.campbell@nyulangone.org (K.A.C.); 4Department of Biomedical Engineering, New York University Tandon School of Engineering, New York, NY 10010, USA

**Keywords:** osteochondral defect repair, CRLF1, mesenchymal stem cells, chondrogenic differentiation, chondrocytes

## Abstract

Background: Since cytokine receptor-like factor 1 (CRLF1) has been implicated in tissue regeneration, we hypothesized that CRLF1 released by mesenchymal stem cells can promote the repair of osteochondral defects. Methods: The degree of a femoral osteochondral defect repair in rabbits after intra-articular injections of bone marrow-derived mesenchymal stem cells (BMSCs) that were transduced with empty adeno-associated virus (AAV) or AAV containing CRLF1 was determined by morphological, histological, and micro computer tomography (CT) analyses. The effects of CRLF1 on chondrogenic differentiation of BMSCs or catabolic events of interleukin-1beta-treated chondrocyte cell line TC28a2 were determined by alcian blue staining, gene expression levels of cartilage and catabolic marker genes using real-time PCR analysis, and immunoblot analysis of Smad2/3 and STAT3 signaling. Results: Intra-articular injections of BMSCs overexpressing CRLF1 markedly improved repair of a rabbit femoral osteochondral defect. Overexpression of CRLF1 in BMSCs resulted in the release of a homodimeric CRLF1 complex that stimulated chondrogenic differentiation of BMSCs via enhancing Smad2/3 signaling, whereas the suppression of CRLF1 expression inhibited chondrogenic differentiation. In addition, CRLF1 inhibited catabolic events in TC28a2 cells cultured in an inflammatory environment, while a heterodimeric complex of CRLF1 and cardiotrophin-like Cytokine (CLC) stimulated catabolic events via STAT3 activation. Conclusion: A homodimeric CRLF1 complex released by BMSCs enhanced the repair of osteochondral defects via the inhibition of catabolic events in chondrocytes and the stimulation of chondrogenic differentiation of precursor cells.

## 1. Introduction

Cartilage injuries of the knee are frequently associated with pain, diminished joint functionality, and reduced life quality. Various surgical techniques to repair or regenerate cartilage injuries are used, including microfracture, cell-based restorative techniques, and osteochondral allografts and autografts [1,2]. Despite the advancement of these surgical techniques, full-thickness chondral or osteochondral defects still remain a major challenge, because none of the surgical methods, including the cell-based approaches, form native hyaline cartilage. As a consequence, the poor quality of regenerated cartilage in the defect site leads to the early onset of post-traumatic osteoarthritis (PTOA) requiring eventual joint replacement at relatively young age. There is an urgent need for new ways to repair cartilage defects or lesions to prevent or delay the onset of early PTOA. 

Cytokine receptor-like factor 1 (CLRF1) belongs to the interleukin-6 (IL-6) superfamily or the gp130 cytokine family [3,4]. CRLF1, as well as the other IL-6 cytokines, can drive both regenerative and degenerative processes via selective and context/cell-specific activation of various signaling pathways [3,4,5]. CRLF1 is expressed in bone and cartilage, and increased expression of CRLF1 in early-stage OA cartilage has been demonstrated [3,6,7,8]. Human mutations in CRLF1 are associated with Crisponi/cold sweating syndromes and lead to early neonatal death in mice due to a suckling defect [3,9]. CRLF1 can form a heterodimeric complex with another member of the IL-6 superfamily, cardiotrophin-like cytokine (CLC) [3,4]. The CRLF1/CLC complex recruits another protein, the ciliary neurotrophic factor receptor (CNTFR), resulting in binding to and signaling through the ubiquitously expressed receptor subunit gp130 and LIFR [3,4]. In addition, CRLF1 can also form a homodimeric complex that signals in the absence of CLC [3,10]. More importantly, the homodimeric CRLF1 complex results in different signaling and cellular responses than the CRLF1/CLC heterodimeric complex [3,10]. Previous studies have suggested that the heterodimeric CRLF1/CLC complex signaling often leads to degenerative processes, while homodimeric CRLF1 complex signaling leads to tissue protection and regeneration [3,10]. For example, a previous study revealed that the CRLF1 homodimer shows neuro-protective effects suggesting that CRLF1 may be a therapeutic target in adults with neurodegenerative conditions [3,10]. 

Little, however, is known about the function of CRLF1 in articular cartilage and in cartilage repair. Previous studies have shown that CRLF1 is highly upregulated in osteoarthritic and damaged cartilage, suggesting a role for CRLF1 in osteoarthritis [4,6,7,8]. In addition, it was shown that the CRLF1/CLC complex stimulated proliferation of the chondrogenic cell line ATDC5 while inhibiting the expression of two cartilage marker proteins, type II collagen and aggrecan [6]. In this study, we analyzed the effects of CRLF1 on chondrogenic differentiation of bone marrow-derived mesenchymal stem cells (BMSCs), catabolic events of articular chondrocytes in an inflammatory environment, and ultimately repair of an osteochondral defect in rabbits. Our findings show that intra-articular injections of BMSCs overexpressing CRLF1 markedly improved the repair of osteochondral defects. 

## 2. Materials and Methods

### 2.1. Adeno-Associated Virus (AAV) Containing Full-Length CRLF1

The AAV vector was generated after cloning full-length human *CRLF1* into adeno-associated vector pENN.AAV.CB7.Cl.eGFP.WPRE.rBG (Plasmid# 105542; Addgene, Watertown, MA, USA). Large-scale AAV1 packaging and purification (titer > 1 × 10^13^ GC/mL) was performed by Vigene Biosciences, Inc. (Rockville, MD, USA). 

### 2.2. Cell Culture 

Human BMSCs obtained from RoosterBio, Inc. (Frederick, MD, USA) were maintained for up to two passages. These human BMSCs are well characterized and are able to differentiate into osteoblasts, chondrocytes, and adipocytes. For this study, BMSCs from female and male donors were used. Cells from different donors were not mixed. Human BMSCs were expanded and cultured in RoosterNourish^TM^-MSC medium according to the manufacturer’s instructions. One day before transduction, cells were plated in appropriate seeding density so that cells were 30–50% confluent at the time of transduction. On the day of transduction, ~1 × 10^6^ BMSCs in 3 mL medium were cultured in the presence of 2.0 × 10^13^ genome copies (GC)/mL of adeno-associated virus (AAV). Cells were incubated at 37 °C in a humidified 5% CO_2_ incubator overnight. Then, medium was replaced with fresh medium and cultured until harvesting or analysis. CRLF1-targeting small interfering RNA (siRNA) duplexes and universal scrambled negative control siRNA duplexes were purchased from OriGene Technologies, Inc. (Rockville, MD, USA). BMSCs were transfected with the siRNAs at a final concentration of 10 nM in suspension using siTRAN siRNA transfection reagent from OriGene Technologies according to the manufacturer’s instructions. The expression levels of CRLF1 were determined by analyzing the mRNA levels of CRLF1 using real-time PCR (qPCR) analysis three days after transduction with AAV or transfection with siRNA. The amount of CRLF1 in the culture medium was analyzed by SDS gel electrophoresis and immunoblotting with antibodies specific for CRLF1. The presence of the homodimeric form of CRLF1 was determined by gel electrophoresis under non-reducing conditions and immunoblotting with antibodies specific for CRLF1 and CLC. 

The human chondrocyte cell line TC28a2 was obtained from Millipore Sigma (Burlington, MA, USA) and cultured according the manufacturer’s instructions. Once TC28a2 cells reached confluency, they were switched to serum-free medium for 24 h followed by treatment with 10 ng/mL human interleukin-1beta (IL-1β; R&D Systems, Minneapolis, MN, USA) for 24 h in the absence or presence of recombinant human CRLF1/CLC complex (100 µM; R&D Systems, #115-CL). Since no active recombinant human CRF1 protein is commercially available, we transfected TC28a2 cells with the expression vector pcDNA containing CRLF1 as described by us previously [11] to determine how overexpression of CRLF1 affects TC28a2 cells in the absence or presence of IL-1β. Treatments were started 72 h after transfection.

### 2.3. Pellet Formation and Chondrogenic Induction

Human BMSCs transduced with empty AAV or AAV containing *CRLF1* (AAV-CRLF1) were trypsinized and 4 × 10^5^ cells were centrifuged at 500× *g* for 10 min. Human BMSCs transduced with AAVs or transfected with siRNAs were left in pellet form in the tubes overnight. The culture medium was replaced with chondrogenic medium consisting of Dulbecco’s Modified Eagle’s Medium, High Glucose (4.5 g/L; #11995-065, Gibco, Waltham, MA, USA) containing ITS+ Premix (6.25 µg/mL each of insulin, transferrin, and selenous acid; 1.25 mg/mL bovine serum albumin; and 5.35 µg/mL linoleic acid; Corning, Corning, NY, USA), 50 µg/mL L-ascorbic acid-2-phosphate (Sigma), 0.1 µM dexamethasone (Sigma), and 10 ng/mL transforming growth factor-beta1 (TGF-β1; Peprotech, Rocky Hill, NJ, USA). Cells were cultured for up to 6 days in differentiation medium in pellets with medium being changed every other day. Total RNA was analyzed after 6 days of pellet cultures. In addition, staining with alcian blue (Sigma-Aldrich, St. Louis, MO, USA) in 3% glacial acetic acid solution of sections of the pellets was performed to determine the degree of chondrogenic differentiation.

### 2.4. Reverse Transcription-Polymerase Chain Reaction (RT-qPCR) and Real-Time PCR Analysis

Total RNA was isolated from BMSC or TC28a2 cell cultures using a RNeasy Mini kit (Qiagen, Valencia, CA, USA). Levels of messenger RNA (mRNA) were quantified by real-time PCR (qPCR) as previously described [12]. cDNA was obtained by reverse transcription of 1 µg of total RNA using the High Capacity cDNA synthesis kit (Applied Biosystems, Foster City, CA, USA). A 1:100 dilution of the resulting cDNA was used as the template to quantify the relative content of mRNA of aggrecan, CLC, CRLF1, interleukin-6 (IL-6), matrix metalloproteinase-13 (MMP-13), Sox-9, type II collagen (α1(II)), and type X collagen (α1(X)) by real-time PCR (ABI StepOne Plus; Applied Biosystems) using the appropriate primers listed in Appendix A and RT^2^ SYBR Green ROX FAST Mastermix (Qiagen). Primers were purchased from Integrated DNA Technologies (IDT; Coralville, IA, USA). qPCRs were performed at 95 °C for 10 min, followed by 40 cycles of 95 °C for 10 s and 60 °C for 30 s, and 1 cycle of 95 °C for 15 s and 60 °C for 1 min. We determined that 18S rRNA and ribosomal protein L13a (RPL13a) were the most stable genes to be used as reference genes for RNA normalization in BMSCs and TC28a2 cells. The cycle threshold (Ct) values for 18S and RPL13a RNA and the samples were measured and calculated. Relative transcript levels were calculated as x = 2^−ΔΔCt^, in which ΔΔCt = ΔE − ΔC, ΔE = Ct_exp_ − Ct_(18S+RPL13a)/2,_ and ΔC = Ct_ctl_ − Ct_(18S+RPL13a)/2_.

### 2.5. Sodium Dodecyl Sulfate (SDS)-Polyacrylamide Gel Electrophoresis and Immunoblotting

Medium was collected from cultured BMSCs transduced with empty AAV or AAV containing CRLF1 and combined with 4 x NuPAGE SDS sample buffer without and with reducing agent (Invitrogen, Carlsbad, CA, USA). After denaturation at 70 °C for 10 min, the medium samples were analyzed by electrophoresis in 10% bis-Tris polyacrylamide gels followed by electroblotting onto nitrocellulose filters. After blocking with a solution of low-fat milk protein, blotted proteins were immunostained with primary antibodies specific for β-actin (#3700, Cell Signaling Technology, Danvers, MA, USA), CRLF1 (ab211438, Abcam, Boston, MA, USA) or CLC (ab154798, Abcam), total STAT3 (#9139, Cell Signaling Technology), phosphorylated STAT3 (#9145, Cell Signaling Technology), total Smad2 and 3, phosphorylated Smad 2, and phosphorylated Smad3 (SMAD2/3 Antibody Sampler Kit #12747, Cell Signaling Technology), and then peroxidase-conjugated secondary antibody (Thermo Scientific Pierce, Waltham, MA, USA). The signal was detected by enhanced chemiluminescence (Pierce) as previously described [12].

### 2.6. Femoral Osteochondral Defect Model

This study utilized a validated osteochondral defect model using skeletally mature male New Zealand white rabbits (6–7-month-old, average weight 3.9 kg) as described previously [13,14,15]. Rabbits were allowed to acclimatize for 7 days before osteochondral defect surgery. After anesthesia, a stainless-steel drill was used to create 4 mm diameter and 4 mm deep cylindrical defects in the center of the distal articular cartilage of the femur. After osteochondral defect surgery, rabbits were randomly divided into three treatment groups, with each group having the same number of rabbits (*n* = 5 per group). Group 1 received three intra-articular injections of 1 mg/mL high molecular weight hyaluronan (HMWHA, Orthovisc^®^) in phosphate-buffered saline (PBS); Group 2 received three intra-articular injections of 1 × 10^6^ BMSCs transduced with empty AAV and resuspended in HMWHA/PBS; Group 3 received three intra-articular injections of 1 × 10^6^ BMSCs transduced with AAV-CRLF1 and resuspended in HMWHA/PBS. The first intra-articular injection was performed directly after osteochondral defect surgery followed by the second injection one week after surgery and the third intra-articular injection two weeks after surgery. We used Orthovisc^®^ high molecular weight hyaluronan (HMWHA, 1–3 million Daltons), which was obtained from DePuy Synthes (Johnson & Johnson, Raynham, MA, USA). Orthovisc^®^ HMWHA is approved by the FDA for the treatment of osteoarthritis. Orthovisc^®^ HMWHA is at a concentration of 15 mg/mL. We first diluted Orthovisc^®^ HMWHA to a concentration of 1 mg/mL using sterile PBS. A cell pellet of 1 × 10^6^ BMSCs transduced with empty AAV or AAV containing CRLF1 was then resuspended in 0.5 mL Orthovisc^®^ HMWHA at a concentration of 1 mg/mL in PBS and used for intra-articular injection into the rabbit knee joints of Groups 2 and 3. Orthovisc^®^ HMWHA at a concentration of 1 mg/mL in PBS was used for intra-articular injections into rabbit knee joints of Group 1.

Based on a false discovery rate of 0.005, a power calculation was revealed to select 5 rabbits in each group. This number of animals per group was calculated based on the statistical power for detecting at least >25% difference in the average macroscopic and histological scores caused by intra-articular injections of BMSCs transduced with AAV containing *CRLF1* in HMWHA/PBS compared to intra-articular injections of BMSCs transduced with empty AAV in HMWHA/PBS. Postoperatively, the animals were permitted to move freely within their cages. Animals were euthanized 12 weeks after surgery and the joints were used for the evaluation of the repair. All animal experiments were performed in accordance with the Declaration of Helsinki of the World Medical Association and approved by the Institutional Animal Care and Use Committee (IACUC) of NYU Grossman School of Medicine (Protocol ID: IA17-01292).

### 2.7. Macroscopic Evaluation

Two blinded and independent reviewers assessed the degree of repair of the osteochondral defect using the International Cartilage Repair Society (ICRS) macroscopic evaluation system [16]. The ICRS macroscopic evaluation system scores three features of repair: the degree of defect repair, integration to border zone, and macroscopic appearance, with each feature being scored from 0 to 4, for a maximum possible score of 12 for best repair.

### 2.8. MicroCT Examination

After washing with PBS, rabbit femurs were incubated in PBS containing the ionic contrast agent Hexabrix (40% *v*/*v*, Mallinckrodt, Hazelwood, MO, USA) for 6 h. Hexabrix is a contrast medium with high density for microCT imaging. Specifically, the repulsion of the negative charge of Hexabrix and sulfated glycosaminoglycans (GAGs) in cartilage results in less intense labeling of cartilage with Hexabrix that distinguishes articular cartilage from the surrounding tissue in microCT images [17]. All joints were evaluated in a scanning tube providing a volex size of 10.5 mm and scanned at 55 kV, 181 mA, and 110 min of acquisition time using a Skyscan 1172 (Bruker, Billerica, MA, USA). During scanning, the samples were wrapped in paper soaked in PBS to avoid dehydration. Amira software 2023.2 (Thermo Scientific, Waltham, MA, USA) was used to reconstruct femur joints from microCT data based on differential density of bone and Hexabrix-treated cartilage.

### 2.9. Histological and Immunohistochemical Defect Repair Scoring

For histological analyses, the distal parts of the femur were removed, trimmed, and fixed in 4% formalin followed by decalcification with 15% ethylenediaminetetraacetic acid (EDTA) *w*/*v* in PBS. After embedding in paraffin, the decalcified specimens were cut to 5 µm sections. Sections through the center of repaired defects were stained with safranin O or prepared for immunohistochemistry. Safranin O-stained sections were evaluated by two blinded and independent reviewers for the degree of repair using a grading scale adopted from the International Cartilage Repair Society (ICRS) Visual Histological Assessment Scale, consisting of six categories with a total score ranging from 0 to 13 points [18]. In addition, we semi-quantitatively scored the intensity of immunostaining for type I and type II collagen using a grading scale consisting of 4 categories (abundant, moderate, slight, and no staining) with a total score ranging from 0 to 3. Both scores were combined, resulting in a total score from 0 to 19 (Appendix A). Data are expressed as total summed average score from the two reviewers. For immunohistochemical staining, the deparaffinized sections were treated with 3% *v*/*v* hydrogen peroxide to block endogenous peroxidase activity. Sections were pre-treated with Proteinase K solution at 37 °C for 15 min followed by incubation with 10% horse serum in PBS *v*/*v* at room temperature for 30 min, to reduce non-specific staining. Sections were then incubated overnight with mouse anti-type I collagen monoclonal antibody (Abcam, ab6308) at a 1:200 dilution, or mouse anti-type II collagen monoclonal antibody (Invitrogen, 2B1.5. MA1-37493) at 1:200 dilution in PBS containing 0.1% *w*/*v* BSA at 4 °C followed by incubation with biotinylated universal immunoglobulin secondary antibody (Vectastain Elite ABC System) at 1:500 dilution for 30 min at room temperature. Sections were then stained by incubation with 20 mg diaminobenzidine and 5 mL hydrogen peroxide (30%) in 100 mL PBS for 5 min at room temperature.

### 2.10. Statistical Analysis

Data are expressed as means ± standard deviation (SD). To test differences in group means, we analyzed continuous data by a parametric test based on unpaired two-tailed *t* test or two-way analysis of variance (ANOVA) followed by Dunnett’s or Tukey’s post hoc test. Non-continuous data were analyzed by a non-parametric test based on Mann–Whitney U test with Bonferroni correction. For non-parametric analysis of multigroup comparison, Kruskal–Wallis test followed by Mann–Whitney U test was used. To correct for multiple comparisons between the groups, we used a Bonferroni correction and targeted a false discovery rate of 0.05/10 = 0.005. Statistical analyses were performed using GraphPad Prism, version 10.0. In all groups, *p* values < 0.05 indicated statistical significance. Cell cultures and animals were randomly assigned to each experimental group and all samples were evaluated in a blinded manner.

## 3. Results

### 3.1. BMSCs Overexpressing CRLF1 Improve Osteochondral Repair

We utilized a validated rabbit osteochondral defect model [13,14,15]. After the surgical generation of the osteochondral defect, rabbits received three intra-articular injections of HMWHA in PBS, BMSCs transduced with empty AAV (BMSC) resuspended in HMWHA/PBS, or BMSCs transduced with AAV containing *CRLF1* (BMSC-CRLF1) immediately after surgery and 1 and 2 weeks after surgery. Rabbits were euthanized at 12 weeks post-surgery. The degree of repair was analyzed by morphological and histological analyses using validated scoring systems [16,18].

#### 3.1.1. Gross Observations

At week 12 postoperatively, cartilage-like repair tissue that had integrated with the original tissue has filled the osteochondral defect in the rabbit knee joints injected with BMSCs transduced with AAV-CRLF1 mixed in HMWHA/PBS (BMSC-CRLF1), while small clefts and fissures were observed in knee joints injected with BMSCs transduced with empty AAV mixed in HMWHA/PBS (BMSCsAAV). The macroscopic appearance of the repair cartilage of BMSC-CRLF1-injected knee joints was mostly smooth with some minor fibrillations, while the macroscopic appearance of the repair cartilage of BMSCsAAV-injected knee joints was more fibrillated, with areas showing small cracks and scattered fissures. The repair cartilage of HMWHA-injected knee joints showed more areas with cracks and scattered fissures than BMSC-injected knee joints (Figure 1A). The overall mean score at week 12 revealed grade II or nearly normal healing (score 11–8) of osteochondral defects in BMSCs-CRLF1-injected knee joints, between grade III (score 11–8) and grade II (score 7–4) healing of osteochondral defects in BMSCsAAV-injected knee joints, and mostly grade III or abnormal healing (score 7–4) of osteochondral defects in HMWHA-injected knee joints (Figure 1B). 

#### 3.1.2. Histological Analysis

Histological analysis of five sections 50 µm apart of the repair site 12 weeks post-surgery revealed that the defect site of knee joints injected with BMSC-CRLF1 was filled with hyaline cartilage that was nicely stained with safranin O, while the reparative tissue in femurs of rabbits injected with BMSCsAAV showed a more fibrotic appearance with less intense or no safranin O staining, especially in the upper areas of the repair cartilage. In addition, knee joints injected with BMSCs-CRLF1 showed the most improved repair of the subchondral bone in the defect site, as indicated by the lack of safranin O staining and the presence of a trabecular bone structure. In contrast, intense safranin O staining was evident in the subchondral area of the repair site of BMSCsAAV-injected knee joints, revealing the presence of cartilaginous repair tissue in the subchondral bone defect. The repair tissue in femurs of rabbits injected with HMWHA showed mostly safranin O-stained fibrotic tissue in the cartilage repair region and extensive cartilaginous tissue in the subchondral repair region. The repair tissue of osteochondral defects in femurs of rabbits injected with BMSCs-CRLF1 showed a better integration with the original tissue, while BMSCsAAV-injected knee joints showed more obvious clefts and defects (Figure 2A). 

Immunohistochemical staining for type I and type II collagen confirmed that the repair cartilage tissue of BMSCs-CRLF1-injected knee joints showed native hyaline cartilage appearance since the repair cartilage tissue showed only immunostaining for type II collagen but not for type I collagen. In contrast, the repair tissue of BMSCsAAV-injected knee joints showed immunostaining for type I and type II collagen, especially in the upper repair cartilage regions that did not stain with safranin O, indicative of fibro-cartilage (Figure 2B). HMWHA-injected knee joints showed intense immunostaining of type I and type II collagen in the entire repair tissue, indicative of fibrotic repair of the osteochondral defect (Figure 2B). Consequently, the repair score was significantly higher in BMSCs-CRLF1-imjected knee joints than BMSCsAAV-injected knee joints. HMWHA-injected knee joints had the lowest repair score (Figure 2C).

MicroCT reconstruction images of Hexabrix-stained knee joints showed the most improved subchondral bone regeneration in knee joints injected with BMSCs-CRLF1 at 12 weeks compared to knee joints injected with BMSCsAAV- or HMWHA-injected knee joints. While microCT images of the repair region of BMSCs-CRLF1-injected knee joints showed a subchondral bone structure similar to areas adjacent to the repair site, microCT images of the repair sites of BMSCsAAV-injected or HMWHA-injected knee joints showed less well-repaired subchondral bone with remaining cartilage-like structure. The most cartilage-like structure was evident in microCT images of the repair site of knee joints injected with HMWHA (Figure 2D).

### 3.2. Effect of CRLF1 on Chondrogenesis

Human BMSCs, when cultured in a pellet culture system in chondrogenic differentiation medium, underwent chondrogenic differentiation as indicated by increased glycosaminoglycan productions as determined by alcian blue staining of histological sections of these pellets. Reducing CRLF1 expression using siRNA in BMSC pellet cultures in chondrogenic medium resulted in markedly decreased alcian blue staining compared to pellets that were transfected with control siRNA (Figure 3A). In contrast, pellets of BMSCs overexpressing CRLF1 after transduction with AAV containing *CRLF1* showed markedly enhanced alcian blue staining compared to the pellet of BMSCs transduced with empty AAV, revealing an increased rate of chondrogenic differentiation (Figure 3A). Chondrogenic differentiation of human BMSCs in pellet cultures was performed in the presence of TGF-β1. TGF-β1 activated Smad2 and Smad3 signaling as revealed by immuno-positive bands of the phosphorylated forms of Smad2 (p-Smad2) and Smad3 (p-Smad3) on immunoblots of lysates from BMSCs (Figure 3B). More intense immuno-positive bands for p-Smad2 and p-Smad3 were present on the immunoblot of lysates from BMSCs transduced with AAV-CRLF1 compared to lysates from BMSCs transduced with empty AAV (Figure 3B).

Chondrogenic markers, such as type II collagen, aggrecan, and Sox-9 mRNA levels were markedly higher in BMSC pellets transduced with AAV-CRLF1 than in BMSC pellets transduced with empty AAV when cultured for 6 days in chondrogenic differentiation medium, whereas mRNA levels of hypertrophic markers (MMP-13, type X collagen) were decreased in BMSC pellets transduced with AAV-CRLF1 compared to BMSC pellets transduced with empty AAV (Figure 4A). On the contrary, transfection of BMSC pellets with siRNA specific for *CRLF1* (siCRLF1) markedly reduced the mRNA levels of aggrecan, type II collagen, and Sox-9 after 6 days culture in chondrogenic differentiation medium compared to BMSC pellets transfected with control siRNA (siCon). mRNA levels of the hypertrophic markers, type X collagen and MMP-13, were also reduced in BMSC pellets transfected with siCRLF1 compared to the mRNA levels of these markers in BMSC pellets transfected with siCon (Figure 4B). These findings reveal that CRLF1 stimulates chondrogenic differentiation, while inhibiting hypertrophic differentiation.

### 3.3. CRLF1 Forms Homodimers When Overexpressed in BMSCs

CRLF1 mRNA levels in BMSCs transduced with AAV-CRLF1 were markedly increased compared to the levels in BMSCs transduced with empty AAV. Overexpression of CRLF1 did not increase the mRNA levels of CLC (Figure 5A). Immunoblot analysis using antibodies specific for CRLF1 of the culture medium revealed a markedly more intense band for CRLF1 in the medium from BMSCs transfected with AAV-CRLF1 compared to the intensity of the CRLF1 band in the medium from BMSCs transfected with empty AAV (Figure 5B). Under non-reducing conditions, a CRLF-1 immuno-positive band was identified with a molecular weight of ~110 kDa in the medium from BMSCs transfected with AAV-CRLF1 (Figure 5C). Upon reduction, we detected a 55 kDa CRLF1 band, which is consistent with monomeric CRLF1 (Figure 5C). The 110 kDa band in non-reducing gels was not detectable with antibodies specific for CLC, confirming that the 110 kDa band represents the homodimeric form of CRLF1 and not a heterodimeric CRLF1/CLC form. A 22 kDa band corresponding to the CLC monomeric form was detectable in the immunoblots of gels run under non-reducing and reducing conditions with antibodies specific for CLC (Figure 5C).

### 3.4. Effect of CRLF1 and CRLF1/CLC on Chondrocytes in an Inflammatory Environment

To determine how CRLF1 or the CRLF1/CLC complex affects chondrocytes in an inflammatory environment, we cultured human chondrocyte cell line TC28a2 in the absence of presence of IL-1β and the recombinant human CRLF1/CLC complex. Since recombinant human CRLF1 is not commercially available, we overexpressed CRLF1 in TC28a2 cells by transfecting these cells with the expression vector pcDNA containing *CRLF1* followed by treatment with IL-1β for 24 h. Treatment of TC28a2 cells with the CRLF1/CLC complex resulted in increased mRNA levels of IL-6 and MMP-13, whereas overexpression of CRLF1 in TC28a2 cells resulted in decreased mRNA levels of IL-6 and MMP-13 compared to untreated cells or cells transfected with empty pcDNA vector. IL-1β treatment also increased the mRNA levels of IL-6 and MMP-13. The CRLF1/CLC complex further increased the mRNA levels of IL-6 and MMP-13 in IL-1β-treated TC28a2 cells, whereas TC28a2 cells that overexpressed CRLF1 showed reduced mRNA levels of IL-6 and MMP-13 in the presence of IL-1β (Figure 6A). Treatment of cells with the CRLF1/CLC heterodimeric complex resulted in the phosphorylation of STAT3, a primary effector of signaling by this complex [3]. STAT3 signaling is considered one of the major catabolic signaling pathways in osteoarthritis [19]. In contrast, TC28a2 cells that overexpressed CRLF1 did not show an increased level of phosphorylated STAT3 (Figure 6B).

## 4. Discussion

CRLF1, as well as the other IL-6 cytokines, can drive both regenerative and degenerative outcomes via selective and context/cell-specific activation of various signaling modules within the gp130 receptor in various tissues [3,5]. CRLF1 is expressed in bone and cartilage, and several studies have shown increased expression of CRLF1 in early-stage OA cartilage [3,5,6,7,16]. However, the exact role of CRLF1 in cartilage homeostasis, regeneration, and pathology is not known. In this study, we show that overexpression of CRLF1 enhanced chondrogenic differentiation of human BMSCs, while inhibition of CRLF1 expression inhibited chondrogenic differentiation. In addition, when CRLF1 was overexpressed in the chondrocyte cell line TC28a2, catabolic events stimulated by IL-1β were reduced. In contrast, when treated with the CRLF1/CLC heterodimeric complex, catabolic events were increased in TC28a2 cells in the absence or presence of IL-1β. These findings suggest that CRLF1 uses different signaling pathways that have different effects on the phenotype and function of mesenchymal stem cells and chondrocytes.

Intra-articular injections of BMSCs that overexpress CRLF1 resulted in an improved repair of an osteochondral defect in rabbits compared to injections of BMSCs that do not overexpress CRLF1. The repair cartilage in the BMSCs-CRLF1 group was hyaline, as indicated by safranin O staining and type II collagen immunostaining, while the repair cartilage in the BMSCsAAV group showed immunostaining for type I and type II collagen, indicative of the presence of fibro-cartilage at the repair site. In addition, the subchondral bone region in the defect site was repaired to a level similar to the subchondral bone areas next to the repair site, whereas cartilaginous repair tissue was still present in the subchondral bone repair site in the BMSCsAAV group. The group of rabbits which did not receive injection of BMSCs showed the worst healing of the three groups with fibrous tissue at the cartilage repair site and the most cartilaginous-like tissue within the subchondral bone repair site. The findings confirm previous studies showing improved osteochondral repair in rabbits after intra-articular injections of BMSCs [20,21]. More importantly, the overall improved repair of osteochondral defects in the presence of BMSCs overexpressing CRLF1 suggests that CRLF1 can promote the repair of cartilage and subchondral bone in osteochondral defects.

Previous studies have shown that CRLF1 forms a homodimeric complex or a heterodimeric complex with CLC, another member of the IL-6 cytokine family [3,9]. Our study shows that the overexpression of CRLF1 in human BMSCs (transduced with AAV containing *CRLF1*) resulted in the release of a homodimeric complex of CRLF1. More importantly, intra-articular injections of BMSCs overexpressing CRLF1 were more effective in improving the repair of an osteochondral defect in rabbit knee joints than intra-articular injections of BMSCs that do not overexpress CRLF1 (transduced with empty AAV), suggesting that the homodimeric form of CRLF1 is more efficient in promoting osteochondral defect repair than the non homodimeric form of CRLF1. Similar to our findings, a previous study showed that only the homodimeric complex of CRLF1 is neuro-protective and protects neurons from oxidative stress [9].

The signaling of the heterodimeric complex of CRLF1 and CLC involves the use of a receptor subunit ciliary neurotrophic factor receptor (CNTFR) [3]. The resulting complex then recruits the ubiquitously expressed receptor subunit gp130 and LIFR activating STAT3 signaling [4]. In this study, we show that STAT3 signaling was activated when the chondrocytic cell line TC28a2 was treated with the heterodimeric CRLF1/CLC complex. When these cells, however, overexpressed CRLF1, STAT3 signaling was not activated, similar to previous findings showing that the homodimeric form of CRLF1 does not activate STAT3 signaling during neuroprotection from oxidative stress [9]. Currently, no receptor which the homodimeric CRLF1 complex binds has been identified. It is, however, also possible that the homodimeric complex of CRLF1 acts as a decoy receptor that neutralizes the heterodimeric CRLF1/CLC complex in the extracellular environment or inside the cell. Future studies need to address whether CRLF1 homodimers bind directly via a cell surface receptor to MSCs and/or chondrocytes, and determine the potential signaling mediated by the homodimeric CRLF1 complex.

Overexpression of CRLF1 in human BMSCs increased chondrogenic differentiation of these cells in the presence of TGFβ1, as revealed of increased alcian blue staining of sections of BMSC pellets and increased mRNA levels of chondrogenic markers, such as aggrecan, type II collagen, and Sox-9, a major transcription factor regulating chondrogenesis [22]. TGFβ-mediated Smad2 and 3 signaling has been shown to play a crucial role in chondrogenic differentiation and articular cartilage repair [23]. In addition, increased Smad2 and 3 signaling during chondrogenesis has been shown to inhibit hypertrophic and terminal differentiation of chondrocytes [23,24]. Our findings showing increased Smad2 and 3 activation, increased chondrogenesis, and decreased hypertrophic differentiation of BMSCs overexpressing CRLF1 compared to BMSCs that do not overexpress CRF1 suggest that CRLF1 may affect chondrogenesis via the regulation of TGFβ/Smad signaling. Previous studies have shown that CRLF1 expression is upregulated by TGFb [6,25,26], suggesting a positive feedback loop during chondrogenic differentiation of BMSCs, in which increased amounts of CRLF1 further stimulate TGFβ-mediated chondrogenic differentiation.

CRLF1 expression has been shown to be increased in osteoarthritic cartilage [4,6,7,27]. In addition, a previous study has shown that the CRLF1/CLC complex stimulates chondrocyte proliferation and suppresses the expression of aggrecan and type II collagen [6]. Our study reveals that a homodimeric complex of CRLF1 stimulated chondrogenic differentiation of human BMSCs and reduced catabolic events in IL-1β-treated chondrocytes, while a heterodimeric complex of CRLF1 and CLC stimulated catabolic events in chondrocytes. Our findings contradict findings from a previous study showing that CRLF1 causes apoptosis in murine BMSCs and inhibits the chondrogenic differentiation of these cells [28]. The study by Xu et al. [28], however, did not determine whether CRLF1 constituted into a homodimeric form or heterodimeric form with CLC in murine BMSC cultures. Therefore, it is plausible to suggest that a heterodimeric complex of CRLF1 and CLC may lead to apoptosis of MSCs and inhibition of their chondrogenic differentiation, while a homodimeric CRLF1 complex leads to the stimulation of chondrogenic differentiation of BMSCs. As shown in other studies [10] and ours, the homodimeric CRLF1 complex results in different signaling than the heterodimeric CRLF1 and CLC complex, ultimately resulting in a different outcome on cell phenotype.

Limitations of our study include the use of a small animal model. The osteochondral defect model in rabbits is well established and validated [13,14,15]. Rabbits, however, have faster skeletal change and bone turnover than other species and humans. Furthermore, rabbits have different gait patterns from those in humans. In addition, even though none of the rabbits in our study experienced any adverse events due to the intra-articular injection of human BMSCs that were transduced with AAV, we did not perform detailed pharmacological and toxicological analyses. Finally, we did not determine whether the injected BMSCs directly contributed to the repair of the osteochondral defect or indirectly contributed via the release of soluble factors, including CRLF1, and extracellular vesicles. A final limitation of this study is the lack of considering sex as a potential variable. In the current study, we only used male rabbits. Before translating our findings into clinical studies, we need to address all the mentioned limitations of the current study.

## 5. Conclusions

Our study shows that overexpression of CRLF1 in BMSCs greatly enhances their effectiveness in improving the repair of osteochondral defects in rabbits compared to BMSCs that do not overexpress CRLF1. Specifically, our findings suggest that the homodimeric form of CRLF1 released by BMSCs overexpressing CRLF1 leads to the stimulation of chondrogenic differentiation of precursor cells and to an inhibition of catabolic events in articular chondrocytes. Both events are crucial for the improvement of the repair of an osteochondral defect. Therefore, our findings suggest that the homodimeric form of CRLF1 may provide a novel therapeutic strategy to improve the repair of osteochondral defects.

## Figures and Tables

**Figure 1 cells-13-00757-f001:**
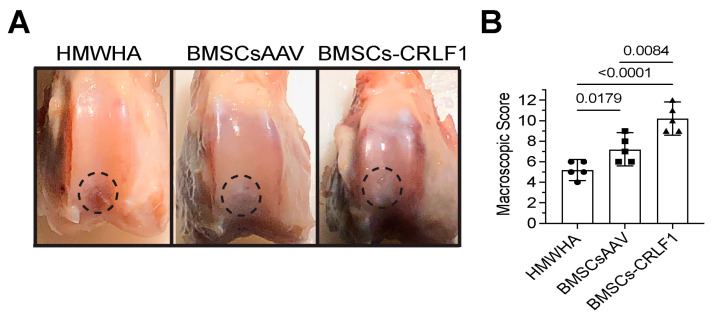
Gross appearance of osteochondral defect repair in rabbit knee joints. (**A**) Gross appearance at 12 weeks post-surgery showing repair of osteochondral defects in a rabbit model after intra-articular injections of high molecular weight hyaluronan (HMWHA; 1 mg/mL in PBS), bone marrow-derived mesenchymal stem cells (BMSCs) transduced with empty adeno-associated virus (AAV) (BMSCsAAV) in HMWHA/PBS, or BMSCs transduced with AAV containing *cytokine receptor-like factor 1* (*CRLF1*) (BMSCs-CRLF1). Dotted circle indicates repair site. (**B**) ICRS macroscopic osteochondral repair assessment for three intra-articular HMWHA injections, BMSCsAAV injections, or BMSCs-CRLF1 injections (*n* = 5 knees per group). Values are mean ± SD. Statistical analysis between the two groups was carried out using Mann–Whitney U test. *p* values are indicated above the graphs.

**Figure 2 cells-13-00757-f002:**
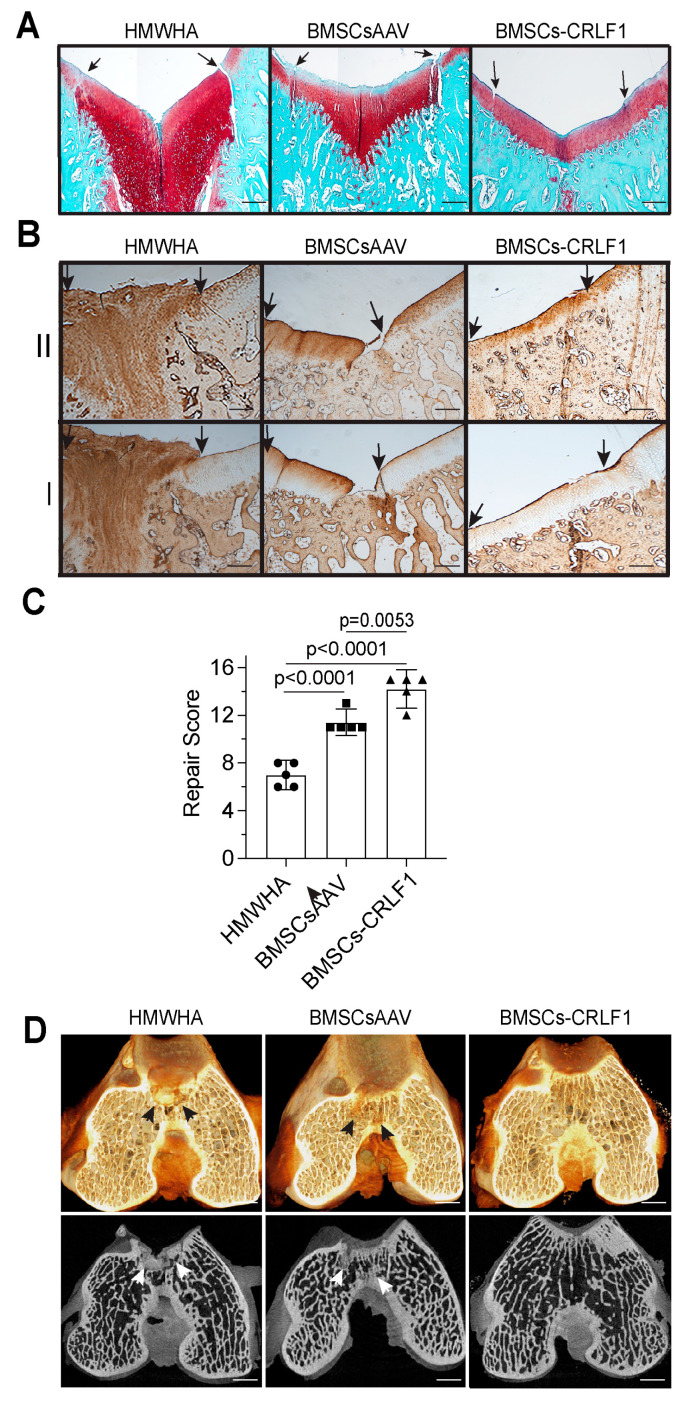
Histological and microCT assessment of the repair of osteochondral defects in rabbit knee joints. Histological and microCT assessment of osteochondral defect repair after three intra-articular injections of HMWHA/PBS, BMSCs transduced with empty AAV in HMWHA/PBS (BMSCsAAV), or BMSCs transduced with AAV containing *CRLF1* (BMSCs-CRLF1) in HMWHA/PBS (*n* = 5 per group) at 12 weeks. (**A**) Safranin O red-stained images of osteochondral repair after HMWHA/PBS, BMSCsAAV, BMSCs-CRLF1 intra-articular injections. Arrows indicate the repair site. Bar, 200 µm. (**B**) Images of sections from osteochondral repair sites after intra-articular injections of HMWHA/PBS, BMSCsAAV, or BMCSsAAV-CRLF1 immunostained with antibodies specific for type II collagen (II) or type I collagen (I). Arrows indicate the repair site. Bar, 200 µm. (**C**) The ICRS Visual Histological Assessment Scale as outlined in Appendix A was used to semi-quantitatively assess the degree of osteochondral repair after intra-articular injections of HMWHA/PBS, BMSCsAAV, or BMSCs-CRLF1 (*n* = 5 per group). Values are expressed as mean ± SD. Statistical analysis between two groups was carried out using Mann–Whitney U test. *p* values are indicated above the graphs. (**D**) Images of three-dimensional microCT reconstructions of Hexabrix-stained osteochondral repair tissue in the HMWHA/PBS-injected, BMSCsAAV-injected, or BMSCs-CRLF1-injected groups. Arrows indicate remaining cartilaginous tissue at the subchondral repair site. Bar, 1 mm.

**Figure 3 cells-13-00757-f003:**
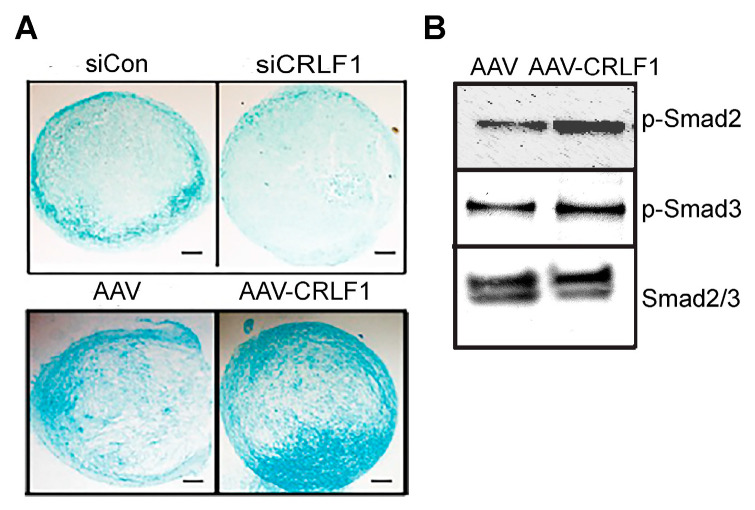
Chondrogenic differentiation of human BMSCs lacking CRLF1 or overexpressing CRLF1. (**A**) Alcian blue staining of sections of BMSC pellets transfected with control siRNA (siCon) or siRNA specific for *CRLF1* (siCRLF1) after 6 days in chondrogenic differentiation medium; sections of BMSC pellet transduced with empty AAV (AAV) or transduced with AAV containing CRLF1 (AAV-CRLF1) after 6 days in chondrogenic differentiation medium. Bar, 100 µm. (**B**) Phosphorylated (p-Smad) and total Smad2 and Smad3 immunoblots of lysates from BMSCs transduced with empty AAV (AAV) or from BMSCs transduced with AAV-CRLF1. Cells were treated for 60 min with transforming growth factor beta 1 (TGFβ1).

**Figure 4 cells-13-00757-f004:**
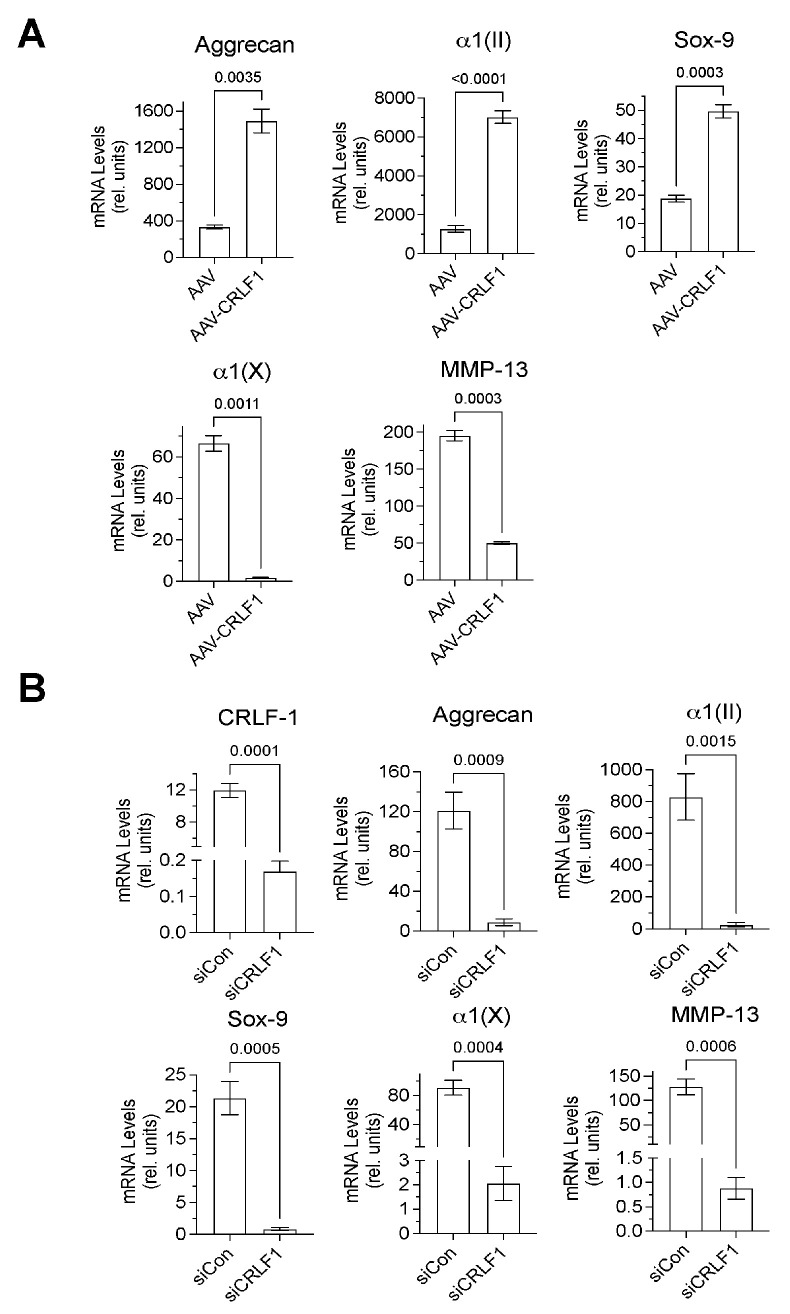
mRNA levels of chondrogenic markers in pellet cultures of BMSCs transduced with empty AAV or AAV containing CRLF1. (**A**) mRNA levels of articular chondrocyte markers (aggrecan, type II collagen α1(II), Sox-9), hypertrophic differentiation markers (type X collagen α1(X), matrix metalloproteinase-13 (MMP-13)) at day 6 of BMSC pellets transduced with empty AAV (AAV) or AAV containing CRLF1 (AAV-CRLF1) and cultured in chondrogenic differentiation medium for 6 days. (**B**) mRNA levels of CRLF1, articular chondrocyte markers (Aggrecan, type II collagen α1(II), Sox9), hypertrophic differentiation markers (type X collagen α1(X), MMP-13) at day 6 of BMSC pellets transfected with control siRNA (siCon) or siRNA specific for CRLF1 (siCRLF1) and cultured in chondrogenic differentiation medium for 6 days. mRNA levels in (**A**,**B**) were determined by qPCR using SYBR Green and normalized to the level of 18S and RPL13a RNA. The mRNA levels are expressed relative to the levels of BMSC pellets transduced with empty AAV or control siRNA cultured for 1 day in chondrogenic differentiation medium, which was set as 1. Data were obtained from triplicate PCRs using RNA from 3 different cultures (*n* = 3), and are expressed as mean ± SD. Statistical analysis between the two groups was carried out using a two-tailed *t* test; *p* values are indicated above the graphs.

**Figure 5 cells-13-00757-f005:**
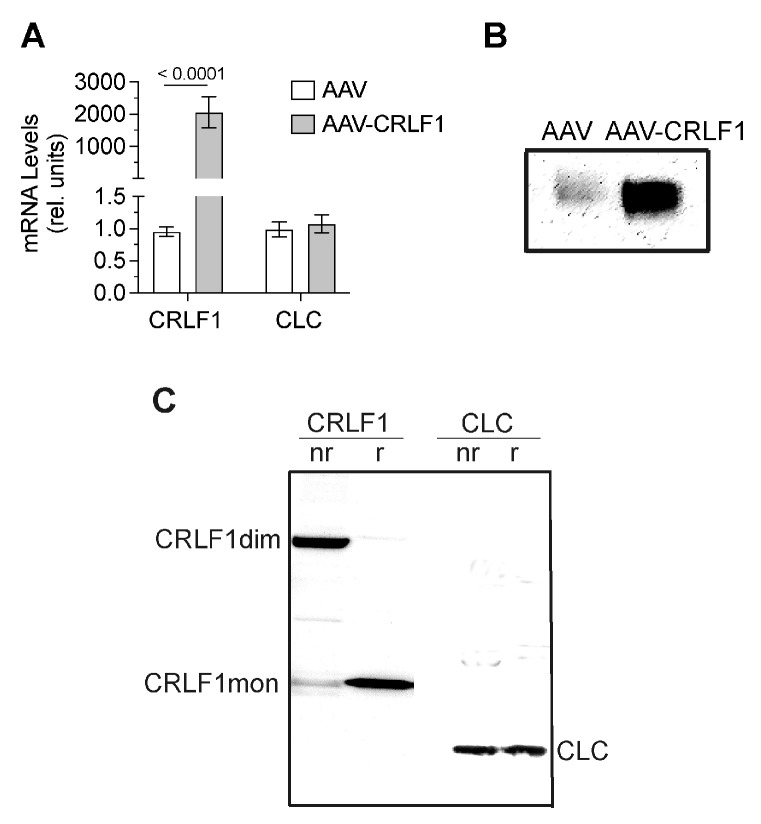
mRNA and protein levels of CRLF1 in human BMSCs after transduction with empty AAV or AAV containing *CRLF1*. (**A**) mRNA levels of CRLF1 and CLC in BMSCs 4 days after transduction with empty AAV (AAV) or AAV containing *CRLF1* (AAV-CRLF1). Levels of mRNA were determined by qPCR using SYBR Green and normalized to the level of 18S and RPL13a RNA. Data were obtained from triplicate PCRs using RNA from 3 different cultures (*n* = 3), and are expressed as mean ± SD. Statistical analysis between the two groups was carried out using two-tailed t test; *p* values are indicated above the graphs. (**B**) Immunoblot analysis of conditioned medium collected from BMSCs transduced with empty AAV (AAV) or AAV-CRLF1 using antibodies specific for CRLF1. (**C**) Immunoblot analysis of conditioned medium collected from BMSCs transduced with AAV-CRLF1 using antibodies specific for CRLF1 (CRLF1) or antibodies specific for CLC (CLC) of SDS gels that were run under non-reducing (nr) and reducing (r) conditions. CRLF1dim indicates the CRLF1 homodimeric form in the non-reducing gel, while CRLF1mon indicates the CRLF1 monomeric form in the reducing gel. CLC indicates the CLC monomeric form.

**Figure 6 cells-13-00757-f006:**
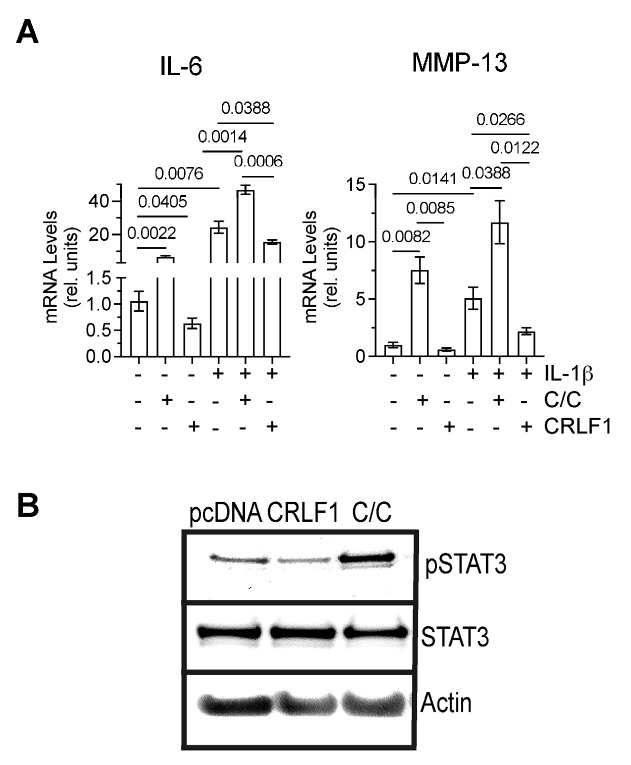
mRNA levels of catabolic markers and STAT3 signaling in TC28a2 cells overexpressing CRLF1 or in the presence of CRLF1/CLC complex. (**A**) mRNA levels of catabolic markers IL-6 and MMP-13 in TC28a2 chondrocyte cell line overexpressing CRLF1 (CRLF1) or in TC28a2 cells treated with the heterodimeric complex of CRLF1 and CLC (C/C) and cultured in the absence or presence of IL-1β for 24 h. mRNA levels were determined by real-time PCR using SYBR Green and normalized to the levels of 18S and RPL13a RNA. The mRNA levels are expressed relative to the levels of TC28a2 cells transfected with empty pcDNA vector, which was set as 1. Data were obtained from triplicate PCRs using RNA from 3 different cultures (*n* = 3), and are expressed as mean ± SD. Statistical analysis between the two groups was carried out using a two-tailed t test; *p* values are indicated above the graphs. (**B**) Phosphorylated (pSTAT3) and total STAT3 immunoblots of lysates from TC28a2 cells transfected with empty pcDNA vector (pcDNA), transfected with pcDNA containing *CRLF1* (CRLF1), or treated with the heterodimeric CRLF1/CLC complex (C/C). Immunoblots for β-actin (Actin) were performed to demonstrate loading of equal protein concentration of each lysate.

## Data Availability

The data presented in this study are available on reasonable request from the corresponding author.

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
