# Peer review of "Cytokine Receptor-like Factor 1 (CRLF1) and Its Role in Osteochondral Repair"

_cells, 2024, doi:10.3390/cells13090757_

Round 1

Reviewer 1 Report

Comments and Suggestions for Authors

The introduction is strong and sets a good foundation for the rest of the paper. The ‘Materials and Methods’ section provides a thorough description of the used experimental methods, including cell culture, AAV generation, chondrogenic induction, animal model creation, etc.; it is a comprehensive and scientifically sound section. The results and their discussion is well-written and insightful. It demonstrates a strong grasp of the scientific concepts and the broader implications of the research. Regarding the command of English language: the entire manuscript is very well-written and flows smoothly. Technical terms are used precisely throughout, and complex ideas and procedures are also conveyed clearly.

Regarding the real time PCR quantification: how much are these experiments in compliance with the MIQE guidelines? According to these, it would be very strongly advisable to 1) test multiple housekeeping gene primer pairs, and 2) adapt the identity of this normalizing control to different experimental settings. (It could even  be the case that the ideal control gene is different for the CRLF1-AAV and the si CRLF1 genes). Please elaborate what were the steps how you adhered to the MIQE guidelines, what is the identity of the normalizing control and what are the limitations. This part is especially important, since the interpretation of a lot of other things – including the transfection and transduction experiments – depend on it. Also, primer sequences should be shared somewhere within the manuscript, or as a supplementary table, at least.

The AAV experiments are well-explained, however, can you please share more details regarding how the siRNA transfections were carried out? Were the cells transfected in pellet form? 70% reduction in CRLF1 levels in the transfected group is a very impressive figure in a pellet setting.

What are the limitations of using the rabbit osteochondral defect model for the in vivo experiments? How well can this study be translated to humans in the future in light of the fact that rabbit articular cartilage has higher intrinsic healing potential and a different level of metabolic activity?

Author Response

Regarding the real time PCR quantification: how much are these experiments in compliance with the MIQE guidelines? According to these, it would be very strongly advisable to 1) test multiple housekeeping gene primer pairs, and 2) adapt the identity of this normalizing control to different experimental settings. (It could even be the case that the ideal control gene is different for the CRLF1-AAV and the si CRLF1 genes). Please elaborate what were the steps how you adhered to the MIQE guidelines, what is the identity of the normalizing control and what are the limitations. This part is especially important, since the interpretation of a lot of other things – including the transfection and transduction experiments – depend on it. Also, primer sequences should be shared somewhere within the manuscript, or as a supplementary table, at least.

 We have described qPCR quantification in more detail and have listed the primer sequences in a table (Supplementary Table 1 in the revised manuscript). We have tested several reference genes and found 18S rRNA and RPL13a are the most stable reference genes for BMSCs and TC28a2 cells.

The AAV experiments are well-explained, however, can you please share more details regarding how the siRNA transfections were carried out? Were the cells transfected in pellet form? 70% reduction in CRLF1 levels in the transfected group is a very impressive figure in a pellet setting.

We have included more details regarding siRNA transfections in the Material and Method section in the revised manuscript.

What are the limitations of using the rabbit osteochondral defect model for the in vivo experiments? How well can this study be translated to humans in the future in light of the fact that rabbit articular cartilage has higher intrinsic healing potential and a different level of metabolic activity?

The limitations of our study are outlined in the last paragraph of the Discussion.

Reviewer 2 Report

Comments and Suggestions for Authors

Author Response

Reviewer 2:

  1. Line 71: Please correct "lime" to the appropriate term.

          Changed to line.

  1. Line 123: Please correct "CRLF1into" to the intended term.

          Changed to CRLF1 into.

  1. Line 165: Kindly provide the IACUC-approved protocol number.

          IACUC Protocol number was added.

  1. Line 190: Please clarify the tissue intended for staining with Hexabrix (neo bone?) and elucidate how Hexabrix stains the tissue of interest.

            The intend of Hexabrix staining and how it stains the tissue is explained in                              Method and Result sections of the revised manuscript.

  1. Line 263: Please confirm whether the sample size is n=5 or n=6.

Sample size is n=5 per group. Sample size of n=5 was confirmed in the text.

  1. Figure 2: Provide a scale bar for histologic images and consider presenting quantification figures for collagen type I and type II staining in Figure 2 (B). Indicate the location of neo bone formation and provide quantification data if feasible for neo bone formation amount in Figure 2 (D).

Scale bars were added to all images. Semiquantitative assessment of type II and I collagen immunostaining is included in the ICRS Visual Histological Assessment scale as outlined in Suppl Table 2. The microCT imagesare from Hexabrix-stained samples. Hexabrix staining allows cartilage to be visible in microCT images. Therefore, we cannot quantitate bone volume since the quantitative analysis would also include remaining cartilage in the subchondral repair site. The microCT images demonstrate the best repair of the subchondral bone region after BMSCs-CRLF1 injections, while cartilage-like structures are still remaining in the repair sites after BMSCsAAV or HMWHA injections. Most cartilage-like tissue remained after HMWHA injections. We have indicated the cartilage like repair tissue at the subchondral bone site in the microCT images with arrows.

  1. Line 322: Confirm whether you are referring to Smad 2/Smad 3 or p-Smad2/p-Smad3.

We confirmed when referring to total or phosphorylated Smad.

  1. In Figure 3(A), could you quantify the amount of glycosaminoglycan?

(Reference

https://www.sciencedirect.com/science/article/pii/S2319417023000665?via%3Di hub for guidance. The mentioned manuscript quantifies GAG by OD570 value in

Figure 8(E).)

We are aware of the quantification of alcian blue staining using guanidine to extract the stain and the quantification of the stain by absorbance OD579nm. Since we used pellet cultures and not micromass cultures as outlined in the referenced manuscript, we had to cut 5µm sections of our pellets. After staining with alcian blue we covered the stained sections with coverslips to protect their integrity and staining. Therefore, it was not possible to extract the stain with a guanidinium solution and quantitate the staining intensity by absorbance at 579nm.

  1. Line 367: Correct "CRLF1band" to the appropriate term.

            We changed to CRLF1 band.

  1. Figures 6A and 6B: Could you provide IL-6 & MMP12 mRNA (A) and pSTAT3 (B) data for control, CRLF1, C/C, IL-1?, IL-1?+CRLF1, and IL-1?+C/C together?

We included a new Figure 6 which includes IL-6 and MMP-13 mRNA levels data for control, CRLF1, C/C, IL-1?, IL-1?+CRLF1, and IL-1?+C/C together (new Figure 6A).

   We did not include pSTAT3 for IL-1ß together with CRLF1 or C/C since the goal of this experiments was to compare the differences of CRLF1/CLC vs CRLF1 on STAT3 signaling, since STAT3 signaling has been shown to be the major signaling pathway of CRLF1/CLC complex.

Reviewer 3 Report

Comments and Suggestions for Authors

The manuscript entitled “Cytokine Receptor Like Factor 1 (CRLF1) and its Role in Osteochondral Repair” is an experimental study involving in vitro cell lines and in vivo experiments. The study is complete, from gene expression to cartilaginous evaluation, exploring pro-inflammatory pathways and chondrogenic differentiation. Below, authors can find some suggestions and concerns.

Abstract is well structured and adequate.

Introduction:

Well structured. Describes the problem, the lack of appropriate treatment and the molecule.

Line 71: cell lime

Methods:

Two distinct BMSCs were used? From a male and a female donors? Were these cultures mixture? Please, clarify.

2.2.: add the incubation time of the cells after transduction and before the CRLF1 gene expression. Most transductions are temporary; thus, the timepoint of gene expression’ analysis is relevant. Moreover, the tc28a2 cell’ section is confusing. Did the authors treat the cells with CRLF1 complex and transfected these cells, or only transfected the cells?

2.4.: as a suggestion, authors could display this section before the 2.2 since the transfection is performed in 2.2. and only described in 2.4.

2.7.: Authors should describe the incorporation of cells into the hyaluronan, and their stability within the gel. Can the authors assure that the cells were alive within the gel before injections? Was the gel Maybe an in vitro assay demonstrating that cells can live within the gel and their release from the gel are viable. Ethical approval number can be added in this section as well. Moreover, the description of sample size calculation is vague, authors could add the alpha, 1 – beta and effect size in a more precise manner.

Results:

One important concern is about the statistics implemented in the present study. Most data are semiquantitative. Therefore, the evaluation of data normality and homoscedasticity can be challenging, precluding the usage of parametric tests.  

Figure 2-B. I and II marks should be on the left. Moreover, the differences in clarity among pictures is substantial. Control is clearly darker than the others. Honestly, in the representative images, the reviewer struggled to see any difference among groups in the area around the arrow.

Figure 2D: Authors pointed out that experimental groups induced more bone formation. However, the reviewer is confusing since calcification of cartilaginous matrix is not a benefit. Moreover, the induction of chondrogenic differentiation should avoid mineralization. Collagen type I is the main protein for bone formation. Thus, as described by the authors, only control groups displayed collagen I-staining in figure 2C. Additionally, the central area of figures 2D seemed to present less cartilaginous tissue in experimental groups in comparison to control (gray area between bone tissue). Please, explain.

Figure 6: Would be important to have the IL6 expression from cells cultured with cc and crlf1 without IL-1B, to see the upregulation of IL6 caused by the experimental compounds. MMP13 as well.

Discussion:

Discussed in a comprehensive manner the dual role of crlf1 in tissue regeneration/ inflammation. However, some discussion in the upregulation of SOX9 (most important transcription factor in chondrogenesis, and other assessed genes), uCT data and histochemical analysis were not included in this section.

Avoid the repetition of “findings”.

Add the ethical approval for animal study, if authors followed ARRIVE guidelines and if the purchased bone marrow cell line needed any ethical approval.  

References:

The reviewer was surprised by the reduced number of references in the present study. Moreover, only 5 references were post 2020. Authors should update them.

Author Response

Reviewer 3:

The manuscript entitled “Cytokine Receptor Like Factor 1 (CRLF1) and its Role in Osteochondral Repair” is an experimental study involving in vitro cell lines and in vivo experiments. The study is complete, from gene expression to cartilaginous evaluation, exploring pro-inflammatory pathways and chondrogenic differentiation. Below, authors can find some suggestions and concerns. 

Abstract is well structured and adequate.

Introduction:

Well structured. Describes the problem, the lack of appropriate treatment and the molecule. 

Line 71: cell lime

Has been changed to cell line.

Methods:

Two distinct BMSCs were used? From a male and a female donors? Were these cultures mixture? Please, clarify. 

Cells from different donors were never mixed. We clarified this issue in the revised Method section.

2.2.: add the incubation time of the cells after transduction and before the CRLF1 gene expression. Most transductions are temporary; thus, the timepoint of gene expression’ analysis is relevant. Moreover, the tc28a2 cell’ section is confusing. Did the authors treat the cells with CRLF1 complex and transfected these cells, or only transfected the cells?

We have included the time points for gene and protein expression analysis in Materials and Methods in the revised manuscript. Regarding the treatment of TC28a2 cells with CRLF1 and CRLF1/CLC complex, we treated TC28a2 cells with human recombinant CRLF1/CLC complex purchased from R&D. Since no active human recombinant CRLF1 protein is commercially available, we decided to overexpress CRLF1 in TC28a2 cells by transfection with pcDNA expression vector. We have clarified this section in the revised manuscript.

2.4.: as a suggestion, authors could display this section before the 2.2 since the transfection is performed in 2.2. and only described in 2.4. 

We did this change as suggested by this Reviewer.

2.7.: Authors should describe the incorporation of cells into the hyaluronan, and their stability within the gel. Can the authors assure that the cells were alive within the gel before injections? Was the gel Maybe an in vitro assay demonstrating that cells can live within the gel and their release from the gel are viable. Ethical approval number can be added in this section as well. Moreover, the description of sample size calculation is vague, authors could add the alpha, 1 – beta and effect size in a more precise manner. 

We used HMWHA at a concentration of 1mg/ml. At this concentration HMWHA is not a gel but a viscous solution in which cells can be easily resuspended.

Results:

One important concern is about the statistics implemented in the present study. Most data are semiquantitative. Therefore, the evaluation of data normality and homoscedasticity can be challenging, precluding the usage of parametric tests.  

Based on the reviewer’s comment, we analyzed continuous data by unpaired two-tailed t-test or one -way analysis of variance (ANOVA). Non-continuous data were analyzed by two-tailed Mann-Whitney U test with Bonferroni correction.

Figure 2-B. I and II marks should be on the left. Moreover, the differences in clarity among pictures is substantial. Control is clearly darker than the others. Honestly, in the representative images, the reviewer struggled to see any difference among groups in the area around the arrow. 

We have put the I and II marks on the left. We apologize for the confusion with the one arrow in Fig. 2B. The arrow indicates the end of the repair region. We have now included a second arrow in the revised figure. These two arrows indicate the repair region. There are clear differences in immunostaining for type I and II collagen in repair cartilage within the three treatment groups.

Figure 2D: Authors pointed out that experimental groups induced more bone formation. However, the reviewer is confusing since calcification of cartilaginous matrix is not a benefit. Moreover, the induction of chondrogenic differentiation should avoid mineralization. Collagen type I is the main protein for bone formation. Thus, as described by the authors, only control groups displayed collagen I-staining in figure 2C. Additionally, the central area of figures 2D seemed to present less cartilaginous tissue in experimental groups in comparison to control (gray area between bone tissue). Please, explain.

We apologize for not explaining the subchondral bone repair well. The histology and microCT images demonstrate that the repair of the subchondral bone region was clearly improved in the BMSCs-CRLF1 injected knee joints than the knee joints of the other two treatment groups. The BMSCsAAV and HMWHA groups still have cartilaginous repair tissue in the subchondral region as indicated by the safranin O staining in the subchondral bone area. Also the microCT images show more cartilaginous tissue left in the repaired subchondral bone in the BMSCsAAV and HMWHA groups than in the BMSCs-CRLF1 group. 

Figure 6: Would be important to have the IL6 expression from cells cultured with cc and crlf1 without IL-1B, to see the upregulation of IL6 caused by the experimental compounds. MMP13 as well. 

New data showing mRNA levels of IL-6 and MMP-13 in the absence of IL-1b are now included in the revised Figure 6.

Discussion: 

Discussed in a comprehensive manner the dual role of crlf1 in tissue regeneration/ inflammation. However, some discussion in the upregulation of SOX9 (most important transcription factor in chondrogenesis, and other assessed genes), uCT data and histochemical analysis were not included in this section. 

We have included some discussion about sox9, and the microCT data in the Discussion.

Avoid the repetition of “findings”.

This was changed accordingly.

Add the ethical approval for animal study, if authors followed ARRIVE guidelines and if the purchased bone marrow cell line needed any ethical approval.  

The protocol number and approval date was added. No ethical approval is required for the commercially available human BMSCs.

References:

The reviewer was surprised by the reduced number of references in the present study. Moreover, only 5 references were post 2020. Authors should update them.

Very little has been published about CRLF1 in the musculoskeletal system. Our reference list includes all the relevant papers published which are relevant to our study. We have added some additional references, especially related to the upregulation of CRLF1 expression by TGFbeta and some other references.

Round 2

Reviewer 2 Report

Comments and Suggestions for Authors

My concerns have been appropriately addressed. The work is suitable for publication now. 

Author Response

Thank you for recommending our manuscript for publication.

Reviewer 3 Report

Comments and Suggestions for Authors

The reviewer congratulates the authors by addressing almost all concerns from the reviewer, improving the description of statistics, methodology and discussion.

Some information regarding cell stability within the gel could be added, as well as a more precise description of the sample size calculation.

Comments on the Quality of English Language

Language is good. Pre-proofing can resolve any minor issue.

Author Response

We have added more information about resuspending BMSCs in HMWHA.

We have provided a more detailed description of sample size calculation.